# Chromosome-Level Genome Assemblies Expand Capabilities of Genomics for Conservation Biology

**DOI:** 10.3390/genes12091336

**Published:** 2021-08-28

**Authors:** Azamat Totikov, Andrey Tomarovsky, Dmitry Prokopov, Aliya Yakupova, Tatiana Bulyonkova, Lorena Derezanin, Dmitry Rasskazov, Walter W. Wolfsberger, Klaus-Peter Koepfli, Taras K. Oleksyk, Sergei Kliver

**Affiliations:** 1Computer Technologies Laboratory, ITMO University, 197101 Saint Petersburg, Russia; a.totickov1@gmail.com (A.T.); andrey.tomarovsky@gmail.com (A.T.); yakupovaar@yandex.ru (A.Y.); klauspeter.koepfli527@gmail.com (K.-P.K.); 2Department of the Diversity and Evolution of Genomes, Institute of Molecular and Cellular Biology SB RAS, 630090 Novosibirsk, Russia; dprokopov@mcb.nsc.ru; 3Laboratory of Mixed Computations, A.P. Ershov Institute of Informatics Systems SB RAS, 630090 Novosibirsk, Russia; ressaure@gmail.com; 4Department of Evolutionary Genetics, Leibniz Institute for Zoo and Wildlife Research (IZW), 10315 Berlin, Germany; derezanin@izw-berlin.de; 5Institute of Cytology and Genetics, Siberian Branch of Russian Academy of Sciences, 630090 Novosibirsk, Russia; rassk@bionet.nsc.ru; 6Department of Biological Sciences, Oakland University, Rochester, MI 48307, USA; wwolfsberger@oakland.edu; 7Department of Biology, Uzhhorod National University, 88000 Uzhhorod, Ukraine; 8Smithsonian-Mason School of Conservation, Front Royal, VA 22630, USA; 9Center for Species Survival, Smithsonian Conservation Biology Institute, National Zoological Park, Washington, DC 20008, USA; 10Biology Department, University of Puerto Rico at Mayagüez, Mayagüez, PR 00682, USA

**Keywords:** genome assemblies, scaffolds, genomes, Hi-C, heterozygosity, mammals, conservation genetics, STR markers

## Abstract

Genome assemblies are in the process of becoming an increasingly important tool for understanding genetic diversity in threatened species. Unfortunately, due to limited budgets typical for the area of conservation biology, genome assemblies of threatened species, when available, tend to be highly fragmented, represented by tens of thousands of scaffolds not assigned to chromosomal locations. The recent advent of high-throughput chromosome conformation capture (Hi-C) enables more contiguous assemblies containing scaffolds spanning the length of entire chromosomes for little additional cost. These inexpensive contiguous assemblies can be generated using Hi-C scaffolding of existing short-read draft assemblies, where N50 of the draft contigs is larger than 0.1% of the estimated genome size and can greatly improve analyses and facilitate visualization of genome-wide features including distribution of genetic diversity in markers along chromosomes or chromosome-length scaffolds. We compared distribution of genetic diversity along chromosomes of eight mammalian species, including six listed as threatened by IUCN, where both draft genome assemblies and newer chromosome-level assemblies were available. The chromosome-level assemblies showed marked improvement in localization and visualization of genetic diversity, especially where the distribution of low heterozygosity across the genomes of threatened species was not uniform.

## 1. Introduction

Each species inhabits a specific environment, a niche, that shapes its unique genome sequence and its expression. Genetic diversity within species is valuable through the existence of the unique combinations of genes and alleles present at a given time in a population, but it is also valuable as it contributes to ongoing evolutionary processes [1]. As environments continuously change, some species can adapt to this change, while others cannot. Understanding factors that determine survival and adaptive potential in response to the environmental change enables a better and effective design of conservation strategies for each species [2,3].

Genome diversity is formed by a balance of mutation, drift, and gene flow contributing to the distribution of diversity across the loci along the chromosomes of individuals in a population. The resulting patterns of variation provide a backdrop for natural selection, enabling adaptation [4,5], and it is generally thought that preserving genetic diversity is required for adaptability: a species that has lost all its reserves of genetic diversity is doomed to extinction [6,7,8,9,10,11,12]. In this context, adaptability is generally understood to depend on the existing genetic variation within each species. Indeed, among many endangered and threatened species, genome-wide genetic diversity has been severely reduced, which is usually seen as a critical sign of vulnerability, as genetically diverse populations should be more resilient to environmental change due to a higher adaptive potential [6,13,14,15].

Heterozygosity has been routinely used to evaluate the genetic potential of a population faced with extinction, and a significant majority of threatened taxa show lower genetic diversity than taxonomically related but not threatened taxa [16]. Historically, genetic diversity has been estimated as heterozygosity across neutral markers without regard to their chromosomal locations [17,18]. Low heterozygosity points to high levels of inbreeding, non-random mating, population fragmentation, and potential recent bottlenecks. This measure is simple and easy to estimate, even from a relatively small number of individuals, if enough loci are examined [18], which is perhaps the main reason why heterozygosity is still widely used in conservation genetics to make estimates of genetic structure, migration rates, and effective population sizes of endangered species [19,20,21].

Conservation biology deals with a huge number of species, but genomic studies of non-model organisms usually have significantly smaller budgets than model organisms used in the biomedical or agricultural sciences, forcing conservation scientists into a continuous trade off between quality and quantity of generated data and its cost. Fortunately, the ongoing reduction in genome sequencing costs gradually allows for the increasingly wider adoption of the whole genome resequencing approaches to estimate genetic diversity among as well as within species. However, this usually requires an existing reference genome assembly of sufficient contiguity and quality to be either available or generated for use in aligning reads and calling variants from the resequenced genomes. The price for generating a quality de novo assembly is still a challenge for most conservation genomics teams, depending on the technology used for genome sequencing (e.g., Armstrong et al., [22]).

A temporary solution to this problem (at least in the short-term perspective) is enabled by the recently developed USD 1000 approach for generation of chromosome-level assemblies from one short-insert Illumina paired-end library and an in situ high-throughput chromosome conformation capture (Hi-C) library [23]. An illustration of this approach can help justify a new path towards future studies of genome-wide patterns of diversity across loci in endangered species.

We collected genetic and genomic data from seven threatened mammalian species for which previous highly fragmented scaffold assemblies and recently generated chromosome-level assemblies (including those generated by the USD $1000 approach) were available. Using these assemblies, we performed a comparison between the analyses based on the traditional genetic data versus the new genomic approach to estimate genetic diversity genome-wide. Our primary objective was to evaluate if the newer, more contiguous assemblies allowed for a better estimation of genetic diversity, localization, and visualization of low heterozygosity regions within genomes.

## 2. Materials and Methods

### 2.1. Genomic Data

Draft and chromosome-level assemblies of eight mammal species were downloaded from the NCBI Genome and DNA Zoo databases (Table 1, Appendix A). Six of the species examined had a total of 19 pairs of chromosomes (2n = 38), one (*Ailurus fulgen*) had 18 pairs (2n = 36), and one (*Bison bison*) had 30 pairs (2n = 60) (Table 1). Short-read libraries were obtained from NCBI SRA [23,24,25,26,27,28]; the corresponding SRA accession IDs are listed per species in Appendix A.

### 2.2. Quality Control and Filtration of Data

Raw data quality control was performed using the *FastQC* [29] and *KrATER v1.1* [30] software. Adapter trimming and filtration by quality was performed in two stages with initial kmer-based trimming of large adapter fragments using the *Cookiecutter* software [31], followed by additional trimming of small fragments and quality filtering using the *Trimmomatic* software, *v0.36* [32].

### 2.3. Alignment and Variant Calling

Filtered reads were aligned to the corresponding reference genome assemblies using the *BWA* tool, v0.7.17 [33]. Read duplicates were marked with the *Samtools* package, v1.9 [34]. Variant calling was performed using bcftools v1.10 [35] with the following parameters: “-d 250 -q 30 -Q 30 --adjust-MQ 50 -a AD, INFO/AD, ADF, INFO/ADF, ADR, INFO/ADR, DP, SP, SCR, INFO/SCR” for *bcftools mpileup* and “-m -v -f GQ,GP” for *bcftools* call. Low quality variants (‘QUAL < 20.0 || FORMAT/SP > 60.0 || FORMAT/DP < 5.0 || FORMAT/GQ < 20.0’) were removed using *bcftools* filter. Finally, variants were filtered by coverage. Only variants in regions with 50–250% of whole genome median coverage were retained.

### 2.4. Identification of X Chromosome, Autosomes, and Pseudoautosomal Region

The position of the pseudoautosomal region (PAR) on the X chromosome was detected in several steps. First, the per-base coverage of the corresponding genome assembly was calculated for each genome library analyzed using the *Genomecov* tool from the *Bedtools* package [36]. Next, median coverage was calculated in stacking windows of 10 kbp. Adjacent windows were merged if their median coverage was at least 70% of the whole genome value, but among the merged windows, only those of 100 kbp or longer were retained. Finally, all the combined windows were merged into final regions if the median coverage of the windows in the gap between them was no lower than 70% of the whole-genome coverage.

Identification of the X chromosome (for all species) and autosomes (for cheetah and red panda) was performed using comparisons of the whole genome alignment (WGA) to the genome assembly (v 9.0) of the reference species (domestic cat, *Felis catus*) and published Zoo-FISH data [37]. The corresponding WGA was generated using *LAST aligner v961* [38]. Synonyms to C-scaffolds of all genome assemblies used in this study are listed in Appendix A.

### 2.5. Comparison of Heterozygosity in Autosomes, X Chromosome, and PAR

For male and female individuals of sea otter and American bison, we compared SNP heterozygosity in autosomes and the PAR in 100 kbp stacking windows using Mann–Whitney nonparametric test. To obtain lesser-greater priors and choose the type of test for comparisons between X chromosome and autosome heterozygosity, we selected five subsets of windows and generated boxplots for them using the Matplotlib library: windows sampled from the (1) whole genomes (all), (2) autosomes only (noX), (3) the X chromosome only (onlyX), (4) X chromosome without PAR (noPAR), and (5) pseudoautosomal region only (PAR). Based on the distribution plot, we chose one-sided tests for both PAR versus autosomes and autosomes versus noPAR with following alternative hypotheses: “PAR is more heterozygous than autosomes” for the first comparison, and “autosomes are more heterozygous than noPAR” for the second. The first comparison resulted in raw *p*-values of 4.9 × 10^−10^ for female (SRR8588177) and 1.1 × 10^−9^ for male (SRR8588180) American bison, and 1.5 × 10^−8^, 3.4 × 10^−14^, and 2.7 × 10^−13^ for female (SRR8597300), male 1 (SRR5768046) and male 2 (SRR5768052) sea otters, respectively. The second test was performed only for females and showed a raw *p*-value 2.4 × 10^−81^ for the female sea otter and a much lower value for female American bison. Even with the Bonferroni correction for multiple comparisons, *p*-values were below the significance level of 0.01, resulting in the acceptance of alternative hypotheses in all cases.

### 2.6. Heterozygosity Visualization

Filtered genetic variants were split into two categories: (1) single nucleotide polymorphisms (SNPs) and (2) insertion-deletions (indels). All subsequent analyses were based on SNPs only. Indels could not be used in this analysis due to the low-quality calls from short reads. Counts of heterozygous SNPs were calculated in non-overlapping windows of 100 kbp and 1 Mbp and scaled to SNPs per kbp. Heatmaps and boxplots were drawn using custom scripts based on the *Matplotlib 2* library [39].

### 2.7. Mapping of Known STR Loci on Chromosome-Level Assemblies of Mustelid Species

Primers of 66 previously published STR loci were extracted from seven different mustelid species— stone marten (*Martes foina),*(stone marten), American marten (*Martes americana*), wolverine (*Gulo gulo*), American badger (*Taxidea taxus*), European badger (*Meles meles*), American mink (*Neovison vison*), and ermine (*Mustela erminea)* [40,41,42,43] and used for in silico PCR using available chromosome-level genome assemblies of six mustelid species (Asian small-clawed otter, sea otter, North American river otter, Eurasian otter, domestic ferret, giant otter) and draft assembly of the American mink (Table 2). First, the in silico PCR was performed using *Simulate_PCR 1.2* [44] where, for the raw amplicons, no more than four mismatches with the target sequence were allowed for each primer, and amplicon length was restricted to 50–1000 bp. Next, additional filtration was performed for each primer pair obtained, with raw amplicons ranked (from minimal to maximal values) by length of amplicon and maximum mismatches in pair—MM score = max (forward primer mismatches, reverse primer mismatches), total number of mismatches (TM score = forward primer mismatches + reverse primer mismatches. The top amplicons were then extracted.

All primer pairs were divided into three categories: NA—no amplification; D—amplified, but failed filtering criteria and declined; and L—localized (passed filtration) (Table 2). The L category contains primers used for further analysis and includes two groups that were selected. In both groups, the top one ranked amplicon was generated from both forward and reverse primers (RF or FR amplicons) and had an MM score of 3 or less. In addition, the requirements for the first group included existence of the only amplicon for primer pair. For the second group, multiple amplicons were allowed but with additional restrictions: either the difference in MM score between the top and adjacent amplicons had to be 2 or higher, or the difference in TM score had to be 3 or higher, respectively. Location of amplicons on C-scaffolds (sensu Lewin et al., [45]) was visualized using custom scripts based on the *Matplotlib 2* library [39].

## 3. Results

### 3.1. Evaluation of the Genome Assemblies

We analyzed the genomes from eight mammal species representing different IUCN Red List categories (Table 1): sea otter (*Enhydra lutris*), cheetah (*Acinonyx jubatus*), clouded leopard (*Neofelis nebulosa*), giant otter (*Pteronura brasiliensis*), red panda (*Ailurus fulgens*), Asian small-clawed otter (*Aonyx cinereus*), American bison (*Bison bison*), and Eurasian river otter (*Lutra lutra*). Each of these species (except Eurasian river otter) was represented by two genome assemblies: the initial draft assembly and a chromosome-level assembly generated from the draft using Hi-C-scaffolding [23].

The draft assemblies were generated using different sequencing and assembly approaches, resulting in assemblies of different lengths and contiguities (Table 1; Appendix A). The scaffold N50 of the draft assemblies ranged from 0.10 Mbp for Asian small-clawed otter to 38.75 Mbp for sea otter (Table 1). The total gap length (Ns, Table 1) also varied considerably among the assemblies, from 1.4 Mbp in giant otter to 195.77 in American bison.

The chromosome-level assemblies included several chromosome-length scaffolds or C-scaffolds [45] that corresponded with the haploid chromosome number (1n) of the species, along with many smaller scaffolds. Between these categories, the lengths differed by orders of magnitude (from kbp to Mbp). The C-scaffolds were ordered according to length, from longest to shortest, without assignment to species-specific karyotype, except for cheetah and red panda for which such an assignment was performed using the Zoo-FISH data and whole genome alignments (see Section 2.3 for details).

Clearly, with the help Hi-C scaffolding, the N50 of the assemblies increased considerably (N50; Chr. vs. Draft, Table 1), as the fragments were aligned in their respective order along the C-scaffolds (N50, Table 1). The most dramatic improvement was observed for the Asian small-clawed otter (×1309) and giant otter (×784), while the smallest was observed for the sea otter (×3.8).

While the contiguity has been remarkably improved, the total gap size in most cases did not increase (Ns, Table 1) except in two of the eight species we considered: by 14.15 Mbp for Asian small-clawed otter, and by 10.49 Mbp in case of giant otter. Unfortunately, the Hi-C data cannot be used to estimate distances between the ordered scaffolds, because Hi-C scaffolding uses a fixed-length stretches of Ns to fill the gaps. For instance, a 500 bp insertion was used in the case of 3D-DNA pipeline for the analyzed chromosome-length assemblies. In the case of *sea otter*, the gap sizes slightly decreased (by 0.74 Mbp), probably due to an extensive correction of misassembles or split on long gaps preceding the scaffolding stage.

### 3.2. Heterozygosity Estimations and Visualization

Heterozygosity is expected to be low in threatened and endangered species [16,19,20,21]. Heterozygosity is clearly diminished across the genomes of some the endangered and threatened species, but there is a difference in how this measure is distributed. The species we analyzed included those well known for low levels of heterozygosity, such as cheetah (Figure 1A) and sea otter (Figure 2A–C), which clearly showed extended regions of low heterozygosity/SNP density across their chromosomes (Figure 2A,B, dark blue). In other species that are also considered to be threatened, such as the Asian small-clawed otter (Figure 1F), red panda (Figure 1E), and American bison (Figure 2D,E), genetic diversity as reflected by higher SNP densities was still present in many chromosomal regions.

The Eurasian river otter is not considered threatened or endangered and has a global LC (least concern) status [46]. However, high heterozygosity in this species was observed only in a few chromosomal regions (Figure 1D), and 1130 Mbp of its genome was much less diverse. Similar levels of heterozygosity were observed in the clouded leopard (0.1 < hetSNPs per kbp 0.75), which is considered vulnerable (VU), and around 800 Mbp of its genome showed extremely low levels of heterozygosity (0.1 hetSNP per kbp), similar to the endangered giant otter. The Eurasian river otter genome assembly was sequenced as a part of the 25 Genomes Project by the Wellcome Sanger Institute, but the origin of the individual sample used was not listed in the SRA database. This example emphasizes the necessity of sequencing several wild individuals of each species in a conservation study before making conclusions about genome-wide heterozygosity.

The distributions of the counts of heterozygous SNPs calculated in non-overlapping windows of 100 kbp and 1 Mbp and scaled to SNPs per kbp are presented in Figure 3. We noted that variant counts between the draft and the chromosome-level assemblies were similar for all species in our analysis (Table 3, number of SNPs). However, representing draft assemblies as density plots is challenging due to the high number of short scaffolds that are generally smaller than the window size of 1 Mb. In a typical contiguous 2.5–3.0 Gbp mammalian genome, the number of 100 kbp windows ranges between 25,000 and 30,000, whereas for a window size of 1 Mbp, the number of windows ranges between 2500 and 3000 (Table 3), which enables easier visualization of SNP density and heterozygosity along C-scaffolds (as in Figure 2A,B). Among the eight studied species, giant otter and Asian small-clawed otter had the smallest scaffold N50 values—0.17 and 0.1 Mbp, respectively (Table 1, Figure 3)—and were the most fragmented among the ones we considered. These two draft genome assemblies also had the smallest numbers of SNPs per 1 Mbp and even per 100 kbp windows (Table 3, Figure 3).

### 3.3. X Chromosome and the Pseudoautosomal Region

C-scaffolds corresponding to the X chromosome were identified in the chromosome-level assemblies using the coverage-based approach and libraries generated from male individuals available for seven of the eight analyzed species. In the case of the Asian small-clawed otter, of which only one female individual was sequenced, the X chromosome was identified from whole genome alignment to domestic cat X chromosome. The depth of coverage counted in 1 Mbp stacking windows for 11 males or females clearly revealed the location of the single pseudoautosomal region on the end of X chromosome as expected (Figure 4). Refinement of PAR borders using 10 kbp windows (see Section 2.3 for details) showed variation in its length among species. The shortest PAR (5.6 Mbp) was observed in cheetah and the longest (7.2 Mbp) in the American bison (Appendix A).

Among the eight species analyzed, whole genome data from both male and female individuals were available only for the sea otter and American bison. For these species, we compared SNP heterozygosity in 100 kbp stacking windows between autosomes and the PAR using the Mann–Whitney nonparametric test. To obtain lesser-greater priors and select the type of test for comparison between X chromosome and autosome heterozygosity, we selected five subsets of windows and generated boxplots for them (Figure 5): windows from the (1) whole genome (all), (2) autosomes only (noX), (3) X chromosome only (onlyX), (4) X chromosome without PAR (noPAR), and (5) pseudoautosomal region only (PAR). For detailed description of comparisons, see Section 2.4. Among tested individuals, we detected statistically significant differences, with heterozygosity of PAR > autosomes (noX) > hemizygous in the male region of X chromosome (noPAR). This pattern is visible in Figure 2D,E for American bison, while for sea otter (Figure 2A–C), it was masked by low heterozygosity and the chosen thresholds for the heatmap.

### 3.4. STR Marker Localization

We localized 66 STR loci on the chromosomal-level genome assemblies of seven mustelid species (Table 2, Appendix A): the sea otter (*E. lutris*), giant otter (*P. brasiliensis*), Asian small-clawed otter (*A. cinereus*), North American river otter *(Lontra canadensis*) and Eurasian *(Lutra lutra*) otter, domestic ferret *(Mustela putorius furo*), and American mink (*Neovison vison*) [40,41,42,43]. Among these mustelid species, only the *N. vison* genome assembly has not yet been scaffolded to chromosome level. Nevertheless, it was included in the analysis to serve as a control for our in silico PCR filtering procedure, as described in Section 2.6, because almost one third of all the STR loci in this analysis was originally developed for this particular species [42,43].

The STR markers in this study came from pre-next generation sequencing publications [40,41,42,43,47]. To start, we tested 20 different American mink STR markers (Appendix A). These were denoted according to the source paper either as *Mvi* [42] or *Mvis* [43]. Among these, 18 were successfully mapped to the American mink genome and those also passed our quality criteria, proving the efficiency of our filtration. One locus (*Mvi1272*) was not found in the assembly, and another (*Mvis022*) did not pass the filtration. We further compared our results in a cross-species validation of 7 ermine (*Mustela erminea*) and seven American mink STR loci, denoted in Appendix A as *Mer* and *Mvi*, respectively [42], using the genomes of North American river otter and *E. lutris*. All seven *Mvi* loci and six out of seven ermine loci (all except *Mer041*) were amplified in vitro. Using the same markers and species, we obtained in silico PCR products for seven mink loci—*Mvis* [43] with only one (*Mvis022*) failing the filtering criteria (Appendix A). At the same time, three out of seven *M. erminea* markers, *Mer030*, *Mer041*, and *Mer082*, did not amplify. One of these, *Mer082*, did not work in the genomes of any of the analyzed species, while *Mer030* was amplified only in Asian small clawed otter and giant otter, but, in either case, did not pass the filtering criteria. However, *Mer041* resulted in an in silico PCR product for American mink, as well as domestic ferret (not tested in Fleming et al. [42]).

In the six species with chromosome-level genome assemblies, approximately half of the STR loci (between 28 and 31, depending on the species) were mapped (localized), while a quarter failed the quality check, and a quarter were not found (Table 2, Appendix A). Overall, the markers originally developed for the American marten (Ma-x), wolverine (Gg-x), and American badger (Tt-x) [41] were the most taxon-specific: the majority of them failed to pass the filtering criteria or did not amplify (Appendix A). In contrast, the American mink-derived markers [40,42,43]) were the most universal for cross-species usage.

By dividing the number of localized markers by the number of chromosomes, we calculated the approximate density of STR markers to be ~1.5 markers per chromosome. Approximately one quarter of the chromosomes (4–6 depending on the species) did not contain any markers (Figure 6, S1–S5—light grey color), and among the labeled chromosomes, the mean density of markers was only ~2 loci per chromosome.

## 4. Discussion

### 4.1. Distribution of Heterozygosity

Genome-wide genetic diversity is usually estimated as heterozygosity—the proportion of sites that contain heterozygous single nucleotide variants across the genome [18]. This yields a single numerical value but does not reveal how variant sites are distributed across the genome, which may be critical for identifying hotspots and cold spots of genetic diversity. A more informative way includes calculation of mean or median heterozygosity in adjacent or overlapping sliding windows of fixed size. The size of the window is a matter of choice depending on the contiguity of the assembly and the questions to be addressed, but a significant part of the genome must be represented in windows to make heterozygosity estimates reliable, especially in the context of runs of homozygosity [48]. For visualization of variant density, a window of 1 Mbp or similar seems to be optimal (Figure 1 and Figure 2), providing clear and easy to understand Figures. However, such window size automatically requires either chromosome-level assemblies or drafts with high N50 to avoid loss of data (Table 3). For the two species with highly fragmented drafts (Asian small clawed otter and giant otter), it was even impossible to perform statistical tests on 1 Mbp windows due to the extremely small numbers of such windows.

Despite significant differences in average genome-wide heterozygosity levels among the species, all eight genomes—of six threatened (three VU and three EN) and two not threatened (one NT and one LC) animals—contained some regions with very low diversity (blue and dark blue regions on Figure 1 and Figure 2). The most striking difference in heterozygosity was observed between different regions in the genome of the giant otter (Figure 2C). Having ~2.5 times higher mean heterozygosity than sea otter [25], the giant otter showed long homozygous stretches (dark blue in Figure 2C) on more than half of its chromosomes.

Assessing and visualizing the distribution of heterozygosity along chromosomes is not the only advantage brought by genomic methods to conservation genetics. Variance in the distribution of diversity and divergence along genomes can be compared between closely related species to detect regions affected by recent and ancient natural selection and introgression [49,50], and annotation of the specific genomic features would help to find specific functional sites where distribution of genomic changes deviates from that expected from the models based on neutral evolution [51]

### 4.2. Mapping Sex Chromosomes and PAR

For both coverage-based and diversity-based methods to work correctly in identifying PARs, chromosome-level assemblies are required, as there is no method to distinguish a decrease in X chromosome-dependent coverage from fluctuations in coverage or decreases in X chromosome-dependent heterozygosity from runs of homozygosity in highly fragmented draft assemblies.

Most of the X chromosome in mammals is hemizygous in males and has a lower diversity in females than that along the autosomes, even after the X/A ratio correction [52]. At the same time, its pseudo-autosomal region (PAR) often shows higher levels of heterozygosity [53,54,55]. Both patterns were observed in the current analysis and were clearly visible on the boxplots (Figure 5) as well as the density maps (Figure 1 and Figure 2). This is because genetic diversity is expected to be higher in the PARs than in the other regions for three different reasons [56]. First, the recombination rate is 5 to 20 times higher in the PAR compared with the genome-wide average [57,58]. If recombination increases the local mutation rate [59,60,61], this will lead to a higher diversity in PARs than in chromosomal regions that do not recombine. Second, recombination can also unlink alleles affected by selection from nearby sites, lessening the effects of background selection and genetic hitchhiking on decreasing genetic diversity [62,63]. Third, diversity should be higher in PARs due to the larger effective population size compared with the nonrecombining regions of the sex chromosomes, because there are two copies of this region present in both males and females. Therefore, pseudoautosomal regions could be found both in males and females. In males, they could be easily mapped comparing the coverage in windows with the whole genome median coverage (Figure 3). In females (with some exceptions; e.g., Cotter et al., [56]), it could be detected also by examining patterns of heterozygosity (Figure 1F, Figure 2A,D). Our findings (Figure 5) suggest that differences between PARs and hemizygous regions of X chromosome can be observed even in such a homozygous species as the sea otter.

The mammalian X and Y chromosomes are commonly excluded from many types of analysis, such as demographic history reconstruction, because of the complexity of inheritance affecting the localization of genetic diversity [55]. This is easy to do with high contiguity, chromosome-level assemblies, because sex chromosomes can be readily detected using a limited number of linked markers, while with more fragmented short-read draft assemblies, the identification of sex chromosomes requires whole genome alignment of scaffolds to the C-scaffolds corresponding to the X and Y chromosomes (if assembled) of related species followed by checking of sequence coverage.

### 4.3. STR Marker Localization

Whole genome sequencing of hundreds of individuals is still too expensive for conservation biology studies, resulting in the common usage of low-resolution methods, such as STR panels, or reduced representation approaches, such as restriction site-associated DNA sequencing (RADseq) [64,65]. The issue with markers such as STR loci, developed in the pre-NGS era, is that they are often not localized on chromosomes, and the relatively small number of loci applied in studies (10–50) is used as a proxy for genome-wide heterozygosity and other assessments. The existing datasets, especially in the historic population studies, can now be merged and compared with the new estimated based on the STR loci that are commonly extracted from the whole genome sequencing in the more recent studies.

Localization of markers on chromosomes is crucially important in studies of interspecific hybridization. It is clear that complex structures like these require labeling of each chromosome, or possibly even each arm of each chromosome, in order to identify and classify hybrids correctly. Our analysis demonstrates that even using a large set (66) of mustelid STR markers developed in the five studies of the pre-genomic era with previously unknown localization, some of the chromosomes were missed in the analysis (Figure 5, chromosomes shaded in grey).

Levels of hybridization, gene flow, and population structure can be very complex, especially where two or more closely related species occur sympatrically. For example, a case of fertile or partly fertile F1 hybrids resulting in backcrosses with parental and maternal species and mosaic F2 hybrids was recently reported for European (*Meles meles*) and Asian (*M. leucurus*) badgers [66]. Both species have 22 pairs of chromosomes (2n = 44), but only 9 microsatellite markers were used in this study. Therefore, 40% of chromosomal linkage groups were included in the analysis, and the remaining 60% were not evaluated. This automatically raises another question: were the individuals reported as F1 by Kinoshita et al. [66] really from the F1 generation, or, alternatively, were some of them F2-s or even backcrosses? Unfortunately, at this point, we do not have a definitive answer to this question due to the lack of data. The absence of chromosomal assemblies clearly diminished the certainty of hybridization studies, as mentioned above in the case of badgers and in the investigation of hybrids between sable (*Martes zibellina*) and pine marten (*M. martes*) [67].

## 5. Conclusions

This study compared highly fragmented draft genome assemblies and recently generated chromosome-level genome assemblies of the eight mammalian species. The analyses of whole genome resequencing data require generation of a reference genome assembly from either the same species or a closely related species. Chromosome-level assemblies can be generated using a combination various long-read or short-read sequencing technologies [23,68]. Inexpensive contiguous assemblies can be generated using Hi-C scaffolding of the existing short-read draft assemblies where N50 of the draft contigs is as low as 0.1% of the estimated length of the genome. Chromosome-length assemblies provide additional benefits, including simplifying the design of STR panels and allowing assessment of previously selected loci. With the help of Hi-C scaffolding, contiguity has been remarkably improved, and we can conclude that chromosome-level genome assemblies provide a more informative way to directly visualize genome-wide genetic diversity. The results of these comparisons illustrate an improvement in representation of genetic diversity, localization, and visualization of heterozygosity across the genomes. The improved understanding of the distribution of heterozygosity and localization of SNP and STR markers afforded by chromosome-level assemblies is particularly applicable for conservation studies of endangered species.

The International Union for Conservation of Nature of Threatened Species now classifies 37,400 species as threatened with extinction in the 2021 edition of the Red List, which is approximately 28% of all assessed species, almost three times the number reported only a decade ago [13,46]. The Earth’s biota may be in the middle of a mass extinction event caused by the adverse impact of anthropogenic activities [69]. At the same time, due to the concerted effort of the genomics community, there is an increased accessibility to chromosome-level assemblies, such as through the Vertebrate Genomes Project consortium, which aims to generate highly contiguous, chromosome-scaffolded assemblies for all ~70,000 vertebrate species using a combination of long-read and Hi-C approaches [68]. Therefore, a comprehensive evaluation of the remaining adaptive potential in endangered species may soon become possible. The application of contiguous chromosome-level assemblies allowed us to localize and visualize low heterozygosity regions within genomes. Applied to the conservation genetics research, this can improve our understanding of the factors contributing to the variation in genome-wide diversity and, hence, potentially help us to devise better evidence-based strategies for endangered species. It will allow us to understand how to design effective conservation strategies [2,3] and, hopefully, avoid the worst-case scenario in conservation biology—species extinction.

## Figures and Tables

**Figure 1 genes-12-01336-f001:**
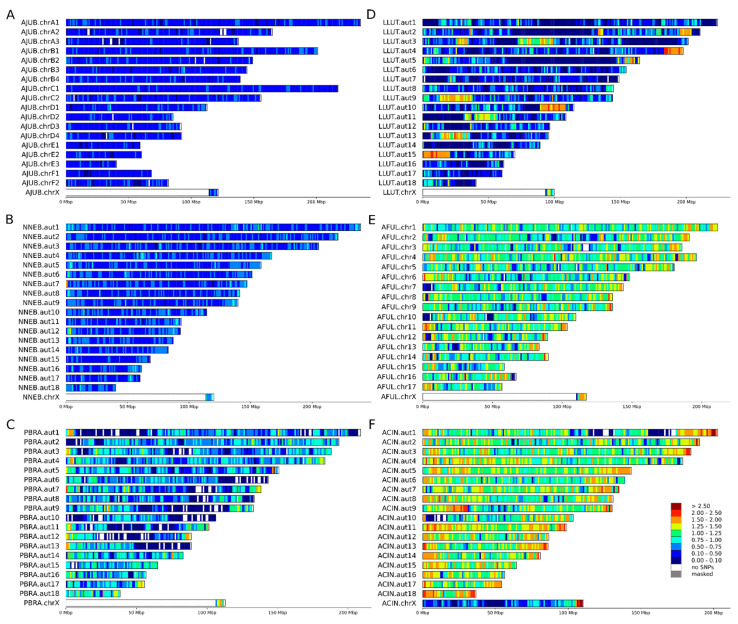
Heatmaps of heterozygous SNP densities for analyzed species based on chromosome-level assemblies for single individuals of six species. Heterozygous SNPs were counted in 1 Mbp windows and scaled to SNP/kbp. (**A**) male cheetah (*Acinonyx jubatus*), (**B**) male clouded leopard (*Neofelis nebulosa*), (**C**) male giant otter (*Pteronura brasiliensis*), (**D**) male Eurasian river otter (*Lutra lutra)*, (**E**) male red panda (*Ailurus fulgens*), (**F**) female Asian small-clawed otter (*Aonyx cinereus*).

**Figure 2 genes-12-01336-f002:**
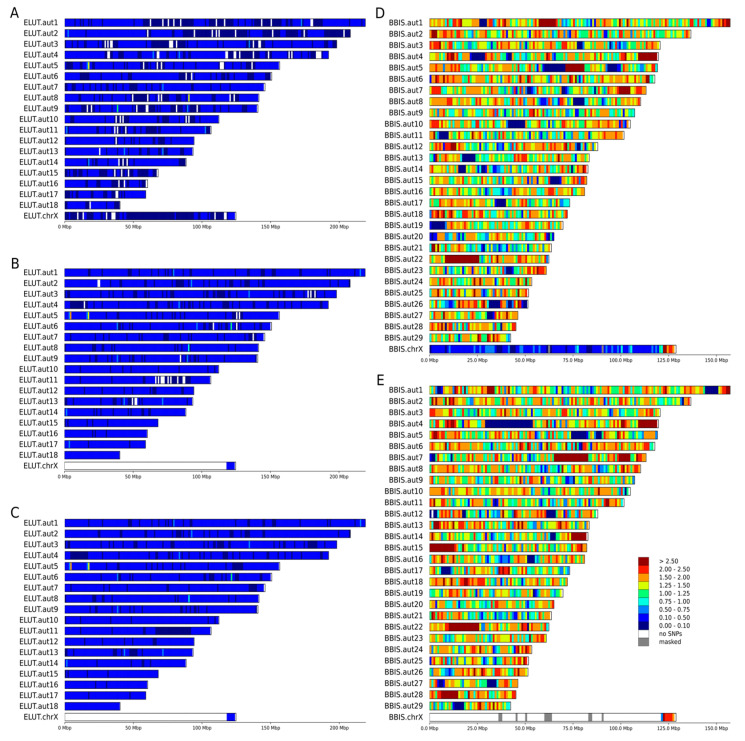
Chromosome-level heatmaps of heterozygosity (density of heterozygous SNP) for two additional species. Individuals of both sexes were available for two the species, sea otters (*Enhydra lutris*) and the American bison (*Bison bison*). Heterozygous SNPs were counted in 1 Mbp windows and scaled to SNP/kbp. (**A**) female sea otter (SRR8597300), (**B**,**C**) male sea otters (SRR5768046, SRR5768052), (**D**) female bison (SRR8588177), (**E**) male bison (SRR8588180).

**Figure 3 genes-12-01336-f003:**
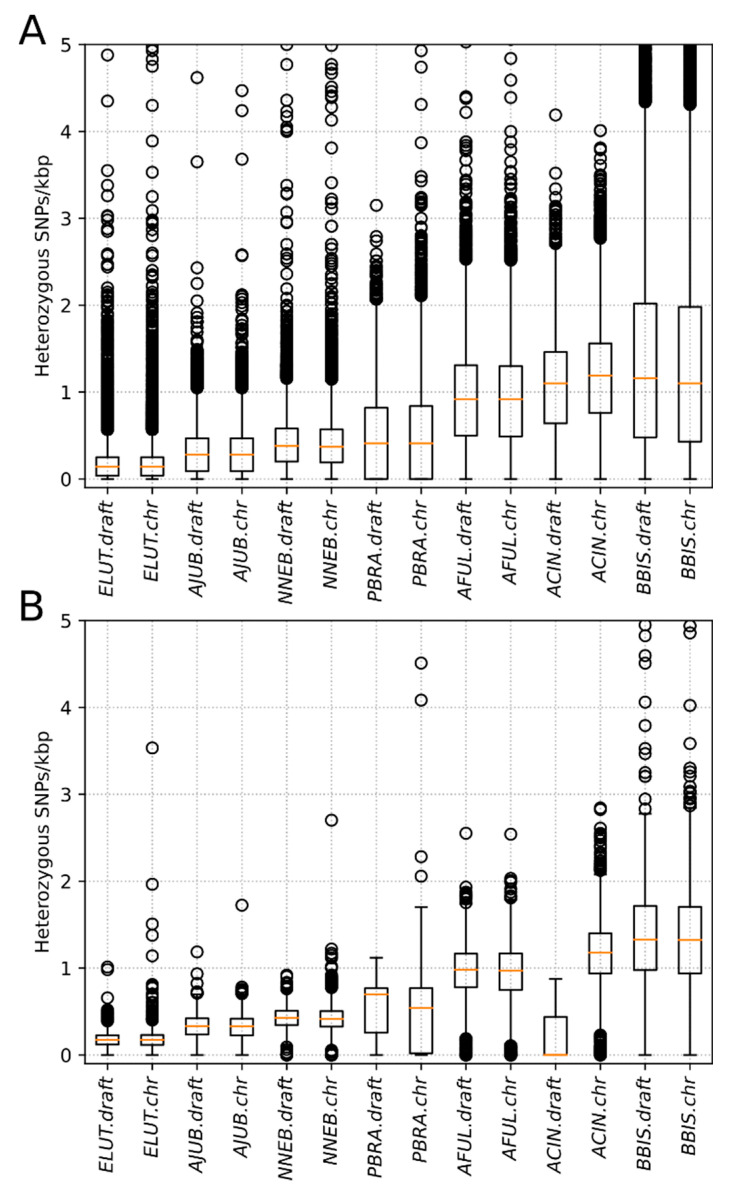
Comparison of the distributions of mean heterozygosity in (**A**) windows of 100 kbp and (**B**) windows of 1 Mbp for the short-read assembled draft genomes and the chromosome-level assemblies. The codes in the Figure are: ELUT—sea otter (*Enhydra lutris*), AJUB—cheetah (*Acinonyx jubatus*), NNEB—clouded leopard (*Neofelis nebulosa*), PBRA—giant otter (*Pteronura brasiliensis*), AFUL—red panda (*Ailurus fulgens*), ACIN—Asian small-clawed otter (*Aonyx cinereus*), and BBIS—American bison (*Bison bison*).

**Figure 4 genes-12-01336-f004:**
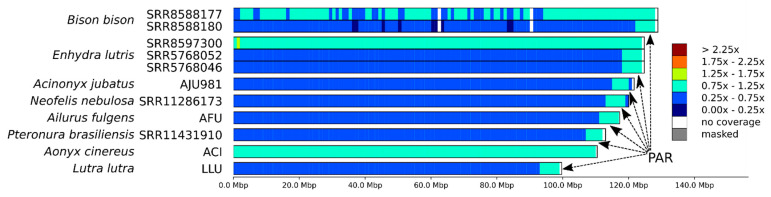
The depth of coverage along the identified X chromosome in eight mammal species (Table 1). The dark blue color corresponds to a half coverage (0.25×–0.75× genome coverage, and the teal-colored fragments are covered at 1× (0.75×–1.25×). Arrows show location of pseudoautosomal region (PAR) in male individuals.

**Figure 5 genes-12-01336-f005:**
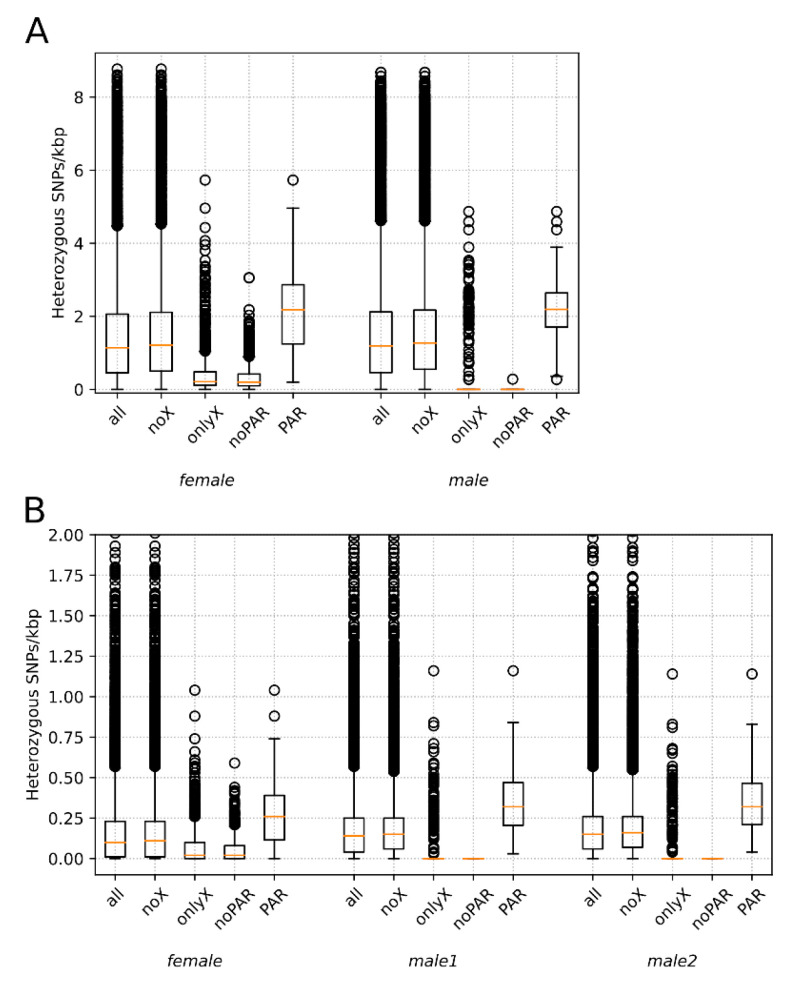
SNP density (SNPs per kbp) in 100 kbp windows inside and outside pseudoautosomal regions in females and males of two mammal species. (**A**) American bisons SRR8588177 (female) and SRR8588180 (male). (**B**) sea otters SRR8597300 (female), SRR5768046 (male 1), and SRR5768052 (male 2). The abbreviations on the *x*-axis stand for the following: all—all 100 kbp windows in genome, noX—all without X chromosome, only X—from X chromosome only, noPAR—from X chromosome without PAR, PAR—from pseudoautosomal region only. For the sea otter, part of outlier windows (with more than 2 SNPs per kbp) is not shown.

**Figure 6 genes-12-01336-f006:**
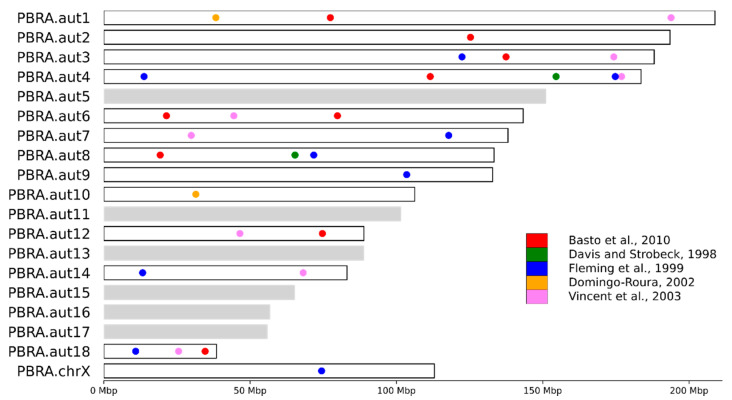
Localization of in silico amplified STR markers on C-scaffolds in the giant otter (*Pteronura brasiliensis*) genome (Basto et al., 2010; Davis and Strobeck., 1998; Fleming et al., 1999; Domingo-Roura, 2002; Vincent et al., 2003) [40,41,42,43,47]. The color of the dots indicates the source publication where marker was developed, the light grey bars show unlabeled (no markers) C-scaffolds. Similar localization maps for other mustelids in this study are shown in Appendix A.

**Table 1 genes-12-01336-t001:** Mammalian species and corresponding genome assemblies used in this study. The measures show the length of the genome size (length), the size of gaps in the assembly (Ns), the N50, and the change in the gap size from draft to chromosome-level assembly (dN).

Species	IUCN Red List Category ^1^	Common Name	2n	Assembly Source or ID	Assembly Level ^2^	Length, Gbp	Ns, Mbp	N50, Mbp	dN, %
*Aonyx cinereus*	VU	Asian small-clawed otter	38	DNA Zoo	Chr	2.44	15.5	130.94	+1048%
				DNA Zoo draft	Draft	2.42	1.35	0.1	
*Enhydra* *lutris*	EN	Sea otter	38	DNA Zoo	Chr	2.45	28.94	145.94	−2%
				GCA_002288905.2	Draft	2.46	29.68	38.75	
*Lutra* *lutra*	LC	Eurasian river otter	38	DNA Zoo	Chr	2.44	0.1	148.99	n/a
*Pteronura brasiliensis*	EN	Giant otter	38	DNA Zoo	Chr	2.46	11.89	133.38	+749%
				DNA Zoo draft	Draft	2.45	1.4	0.17	
*Ailurus* *fugens*	EN	Red panda	36	DNA Zoo	Chr	2.34	34.41	143.8	+1%
				GCA_002007465.1	Draft	2.34	34.04	2.98	
*Acinonyx jubatus*	VU	Cheetah	38	DNA Zoo	Chr	2.37	42.86	144.64	+2%
				GCA_001443585.1	Draft	2.37	42.06	3.12	
*Neofelis* *nebulosa*	VU	Clouded leopard	38	DNA Zoo	Chr	2.42	7.94	147.11	+35%
				DNA Zoo draft	Draft	2.41	5.89	1.38	
*Bison* *bison*	NT	American bison	60	DNA Zoo	Chr	2.83	199.31	101.69	+2%
				GCF_000754665.1	Draft	2.83	195.77	7.19	

^1^ IUCN Red List categories: EN—endangered, VU—vulnerable, NT—near threatened, LC—least concern. ^2^ The assembly levels: draft—initial fragmented assembly, Chr—chromosome-level assembly based on draft and in situ high-throughput chromosome conformation capture (Hi-C).

**Table 2 genes-12-01336-t002:** Mapping of known STR loci onto the six chromosome-level and one draft assemblies of seven mustelid species.

Species	STR Markers			#* Chr**	#* Chr** with Markers	#* Chr** w/o Markers
	Localized (L)	Not Amplified (NA)	Declined (D)			
*Aonyx cinereus* ^1^	31	16	19	19	15	4
*Enhydra lutris* ^2^	26	22	18	19	14	5
*Lontra canadensis* ^3^	28	17	21	19	15	4
*Lutra lutra* ^4^	26	22	18	19	13	6
*Mustela* *putorius furo* ^5^	28	17	21	20	14	6
*Pteronura* *brasiliensis* ^6^	30	18	18	19	13	6
*Neovison vison*^7,^***	36	17	13	15	-	-

^1^—Asian small-clawed otter, ^2^—sea otter, ^3^—North American river otter, ^4^—Eurasian otter, ^5^—domestic ferret, ^6^—giant otter, ^7^—American mink. *—Number of, **—Chromosomes, ***—there is no chromosome-level assembly for American mink available, but we included this species as control. See Section 3.4 for details.

**Table 3 genes-12-01336-t003:** Counts of heterozygous single nucleotide polymorphisms (SNPs), windows and counts, median, and mean heterozygosity in windows of 100 kbp and 1 Mbp for draft and chromosome-level assemblies (Chr.) assemblies of the analyzed genomes. Two species with the lowest window counts are in italic. Bold indicates cases where comparison of mean heterozygosity in windows of 100 kbp and 1 Mbp showed statistically significant difference for significance, level 0.01.

Species	#* Het. SNPs, Millions		Window Size	#*Windows		Median, Het SNPs/kbp		Mean, Het SNPs/kbp		*p*-Value (Draft vs. Chr.)	
	Draft	Chr.		Draft	Chr.	Draft	Chr.	Draft	Chr.	Raw	Adjusted
*Aonyx* *cinereus*	2.73	2.73	100 kbp	9777	22,183	1.100	1.190	1.052	1.144	3.37 × 10^−34^	**2.36 × 10^−33^**
			1 Mbp	*3*	*2204*	*0.001*	*1.177*	*0.292*	*1.146*	*NA*	*NA*
*Enhydra* *lutris*	0.47	0.46	100 kbp	24,146	24,165	0.140	0.140	0.178	0.182	0.98	1
			1 Mbp	2337	2396	0.174	0.176	0.175	0.180	0.79	1
*Pteronura* *brasilensis*	1.25	1.24	100 kbp	13,589	22,819	0.410	0.410	0.488	0.497	0.59	1
			1 Mbp	*32*	*2262*	*0.699*	*0.542*	*0.563*	*0.497*	*NA*	*NA*
*Ailurus* *fulgens*	2.14	2.14	100 kbp	22,083	23,139	0.920	0.920	0.916	0.912	0.50	1
			1 Mbp	1573	2298	0.980	0.971	0.943	0.914	0.17	1
*Acinonyx* *jubatus*	0.75	0.75	100 kbp	22,861	23,609	0.280	0.280	0.314	0.314	0.42	1
			1 Mbp	1757	2350	0.332	0.330	0.322	0.314	0.23	1
*Neofelis* *nebulosa*	1.00	1.00	100 kbp	22,004	23,931	0.380	0.370	0.415	0.407	6.62 × 10^−04^	**0.0046**
			1 Mbp	1194	2387	0.427	0.415	0.426	0.407	1.2 × 10^−03^	**0.0089**
*Bison* *bison*	3.68	3.68	100 kbp	24,286	26,213	1.160	1.100	1.423	1.378	5.33 × 10^−07^	**3.73 × 10^−06^**
			1 Mbp	2181	2604	1.328	1.324	1.414	1.379	0.142	0.9943

*—Number of.

## Data Availability

All data used in this study are publicly available at NCBI as indicated.

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
