# Peer review of "Chromosome-Level Genome Assemblies Expand Capabilities of Genomics for Conservation Biology"

_genes, 2021, doi:10.3390/genes12091336_

Round 1
Reviewer 1 Report
Dear authors,
There are recommendations and modifications (see in pdf file). I suggested you an additional item in material and methods to separate the material biologic from the other analysis. You need to improve the organization of tables, including the title tables. The common names and species names have to be homogenized throughout the text. Discussion can be improved. The conclusions should be improved.

Author Response
Answers to reviewer 1
Line 41. “Remove, this is the title already. Change for mammals, and/or red list , for example”
As recommended we added “mammals” to keyword list and changed “Chromosome-level asemblies” to “genome assemblies”
Line 105-120. Two connected comments
“can you move this parragraph to a new item before quality control called for ex. Biological material. It could be relevant signaled the reference NCBI SRA and associate to the species”
“can you move this parragraph to a new item before quality control called for ex. biological material/ Samples. ”
We added new paragraph “Genomic data” and moved there part of paragraph “ Quality control and filtration of data” as recommended by reviewer. Also we moved SRA ids of libraries to newly created Supplementary table S2 and added association with species. Addition of new supplementary table also shifted numbers of other tables.
Line 123. Caption for Table 1 “specify for the readers how is the N50”
N50 is already described in details in the begging of results as part of comparison of assemblies.
Line 123. First column of table “Species” We renamed first column of Table 1 from “Latin name” to “Species”
Line 130. Only highlighting, but no comment
Obviously version of BWA was missed here. Added “ v0.7.17”
Line 194. Caption for Table 2. “remove and add (Signaled with asterisks). If you prefer add a column to put the common name to the species ”
We removed labeled sentence and added species names as links to table cells
Line 194. Column “2n” in Table 2. “2N is already mentioned in table 1.”
To make Table2 more clearly we replaced column 2n with haploid number of chromosomes (1n). Now it is more compatible with two last columns “#Chr. with markers” and “#Chr. w/o markers”. Also we fixed missing common names for species in lines 182-184.
Table 1 and Table 2 only partly overlap by species. For genetic diversity part of study we used chro-mosome-level assemblies of mammals from different families and even different orders. Main idea here was to include in analysis species of different IUCN categories. For STR-related sections of manuscript we used STRs developed for mustelids and all available chromosome-level assemblies (and draft assembly of American mink for control of in silico PCR) of mustelids.
So manuscript includes three sets of species: (1) species used for genetic diversity analysis, (2) spe-cies with known STR markers, (3) species used to localize of STRs on chromosomes.
All three sets have partial overlaps.
Line 210-217. Two related comments
“I considered that is not a results ! for me, this could go in the section that I proposed added in materi-al and methods : Biological samples”
“add something like :
The eight analyzed mammal genomes showed differences in regard to the genome assembly levels (Draft assemblies and choromosome level assembly).“
We agree with reviewer that from strict point of view on structure of manuscript first sentences of Re-sults section can be moved to methods. But while reading the papers researchers often after a brief look on introduction jump directly to results and return too methods only if something is unclear or if interested in specific protocol. Sentences from lines 210-217 were added to the results to address such pattern of reading.
It could be removed, we leave final decision for editors.
Line 231. “remove bold, idem for the others”.
Bold was removed from all links to tables and figures.
Line 278. Caption for Figure 1.2 “no bold”
Bold was removed.
Line 294 “it could be comment a sentence about P-value difference between Draft vs Chr. window ”
Line 301 Removal of redundant dot
Accepted
Line 304. Removal of some words
Accepted
Line 307. Table 3. 2 related comments
“It is better that species name be in only one line”
“Perhaps if you use only two significant digits you could have more space in this table”
We removed third significant digit from second and third columns as recommended. For other col-umns it makes sense to retain 3 digits as for some species difference between chromosome-level and draft assemblies appear only in 3rd digit. However, it better to leave decision to editors at final proof-reading.
Line 318. “how many?”
It is unclear what does reviewer mean by this comment. Shortest (5.6 Mbp) and longest (7.2 Mbp) PAR lengths are already mentioned in this sentence
Line 323. Caption to Figure 3.
“can you indicate on the Fig.?”
Label PAR and arrows were added to the figure.
Caption was modified in accordance.
Line 341. “verify if you mean male here.”
Reviewer is right. Mistyping was fixed
Line 328-332.
“For me this due go in materials and methods. and for fig. 4 not results are described.”
We believe that figure 1 is a good visual representation that provides an important support to the aim of the paper, and therefore would like to keep it as is
Line 396.
“are they eight mammals??”
Yes, corrected.
Line 397 - 404. “This is a conclusion and not part of the discussion.”
This paragraph was moved to the beginning of conclusion.
Line 410-414
“how about your results?”
We added related text.
Line 424.
“there are some other mammals not threatened to compared your results?”
Yes, in our diversity analysis we included six threatened species - three vulnerable (VU) and three endangered (EN) , and two not threatened (one NT and one LC). We added clarification in corresponding sentence.
Line 500. “look your main objetive!”
To address this comment, a new paragraph was added to the conclusions and the existing text modi-fied to emphasize that the main objective was achieved.
Reviewer 2 Report
I found that the paper was really well-written and easy to follow and that its basic message, while not unexpected or difficult to believe, is an important one. Further, as the entire conservation and population genomics community continues to learn what new methods offer, it is important that even straightforward conclusions be ground-truthed in the way this paper has. So with the proviso that I am not the reviewer to comment on details of the informatics, I am happy to support the paper's publication with a few minor typos corrected and some extra thought to whether gene flow and introgression might be mentioned where I have indicated in the attached pdf

Author Response
Line 53. “and gene flow I guess”
As recommended we added “gene flow”
Line 57. “Perhaps worth looking at some of the papers by Ary A Hoffman et al on this topic. Some are re-veiws and some are primary data papers showing how reduced additive genetic variance can prevent adaptation”
Added citation to Hoffmann, A., Sgrò, C. Climate change and evolutionary adapta-tion. Nature 470, 479–485 (2011). https://doi.org/10.1038/nature09670
Line 397 “help of Hi-C”
added missing word “of”
Line 419. “and introgression I imagine too”
Added “and introgression”
Line 523. Only highlighting, but no comment.
We split highlighted sentence in two.
Round 2
Reviewer 1 Report
This is a better version of this manuscript.